# Cyclodextrin–Amphiphilic Copolymer Supramolecular Assemblies for the Ocular Delivery of Natamycin

**DOI:** 10.3390/nano9050745

**Published:** 2019-05-15

**Authors:** Blanca Lorenzo-Veiga, Hakon Hrafn Sigurdsson, Thorsteinn Loftsson, Carmen Alvarez-Lorenzo

**Affiliations:** 1Faculty of Pharmaceutical Sciences, University of Iceland, Hofsvallagata 53, IS-107 Reykjavik, Iceland; blv3@hi.is (B.L.-V.); hhs@hi.is (H.H.S.); thorstlo@hi.is (T.L.); 2Departamento de Farmacología, Farmacia y Tecnología Farmacéutica, R+D Pharma Group (GI-1645), Facultad de Farmacia and Health Research Institute of Santiago de Compostela (IDIS), Universidade de Santiago de Compostela, 15782 Santiago de Compostela, Spain

**Keywords:** block copolymers, cyclodextrins, ocular drug delivery, fungal keratitis, natamycin, mixed micelles, poly(pseudo)rotaxane, solubility, HET-CAM assay, ocular permeability

## Abstract

Natamycin is the only drug approved for fungal keratitis treatment, but its low water solubility and low ocular penetration limit its efficacy. The purpose of this study was to overcome these limitations by encapsulating the drug in single or mixed micelles and poly(pseudo)rotaxanes. Soluplus and Pluronic P103 dispersions were prepared in 0.9% NaCl and pH 6.4 buffer, with or without α-cyclodextrin (αCD; 10% *w*/*v*), and characterized through particle size, zeta potential, solubilization efficiency, rheological properties, ocular tolerance, in vitro drug diffusion, and ex vivo permeation studies. Soluplus micelles (90–103 nm) and mixed micelles (150–110 nm) were larger than Pluronic P103 ones (16–20 nm), but all showed zeta potentials close to zero. Soluplus, Pluronic P103, and their mixed micelles increased natamycin solubility up to 6.00-fold, 3.27-fold, and 2.77-fold, respectively. Soluplus dispersions and poly(pseudo)rotaxanes exhibited in situ gelling capability, and they transformed into weak gels above 30 °C. All the formulations were non-irritant according to Hen’s Egg Test on the Chorioallantoic Membrane (HET-CAM) assay. Poly(pseudo)rotaxanes facilitated drug accumulation into the cornea and sclera, but led to lower natamycin permeability through the sclera than the corresponding micelles. Poly(pseudo)rotaxanes made from mixed micelles showed intermediate natamycin diffusion coefficients and permeability values between those of Pluronic P103-based and Soluplus-based poly(pseudo)rotaxanes. Therefore, the preparation of mixed micelles may be a useful tool to regulate drug release and enhance ocular permeability.

## 1. Introduction

Fungal keratitis or keratomycosis is a globally distributed ocular infection that can lead to visual impairment, and in the worse cases, blindness. In general, this cornea disease is caused by *Aspergillus* and *Fusarium* spp. in subtropical and tropical regions and by *Candida* spp. in high temperate areas [1,2]. The use of contact lenses, recent ocular trauma, surgery, or treatment with ocular steroids has been identified as risk factors in the development of this infection [3,4,5].

There are three classes of antifungals for the treatment of keratomycosis: polyenes, triazoles, and echinocandins [6]. Natamycin belongs to the group of polyene antifungal antibiotics, and is the only drug approved for the topical ophthalmic treatment of fungal keratitis in the form of suspension eye drops (Figure 1A) [7,8]. Voriconazole has shown higher ocular penetration capacity, but better results have still been reported for patients treated with natamycin. Natamycin suspension is only approved for topical administration [5,7,9]. Other drugs such as amphotericin B, itraconazole, ketoconazole, or fluconazole can be administered by other routes for systemic treatment.

The safety of polyene drugs, such as natamycin, relies on their selective interaction with ergosterol at the fungi wall, avoiding interactions with human cholesterol-containing membranes [10]. Thus, a variety of strategies have been explored to increase natamycin solubility (~50 mg/L in water [11]), stability [12], precorneal residence time, and corneal permeability. Several types of delivery systems such as nanoparticles [13,14,15], hydrogels [16,17], and drug/cyclodextrin complexes [11,18,19,20] have been proposed. Janga et al. [16] designed ion-sensitive in situ gels of natamycin bilosomes for efficient ocular delivery. Bhatta et al. [13] developed lecithin/chitosan nanoparticles to prolong the ocular penetration of natamycin at reduced dose and dosing frequency. Also, Chandasana et al. [14] developed poly-d-glucosamine functionalized polycaprolactone nanoparticles for prolonged drug release. Phan et al. [19] analyzed the release of natamycin loaded-poly(d,l-lactide)-dextran nanoparticles with the purpose of developing drug-eluting contact lenses. Koontz et al. [18] studied the complex formation between natamycin and β-cyclodextrin (βCD), hydroxypropyl-β-cyclodextrin (HPβCD), and γ-cyclodextrin (γCD). They managed to increase drug apparent solubility 16-fold, 73-fold, and 152-fold, respectively, although using quite high CD concentrations.

Despite the growing interest in polymeric nanomicelles as ocular drug carriers, only one reference to their use as natamycin nanocarriers was found. In that previous paper by Loh et al. [21], an in situ gelling derivative of Pluronic was evaluated for sustained release, but the solubilizing capability and ocular application were not considered. Compared to other ophthalmic nanocarriers, polymeric nanomicelles (typical diameter 10–100 nm) require simpler preparation and exhibit mucoadhesion to the ocular surface, better penetration, enhanced stability, and larger cargo capacity [22,23,24,25,26]. Linear triblock poly(ethylene oxide)(PEO)–poly(propylene oxide)(PPO)–poly(ethylene oxide)(PEO) copolymers known as poloxamers or Pluronics^®^ (Figure 1B) are versatile components of nanomicelles with the extra capabilities of inhibiting P-glucoprotein efflux pumps at the eye surface [27] and undergoing sol-gel transitions upon heating [23,28]. Lately, Soluplus^®^, a polyvinyl caprolactam–polyvinyl acetate–polyethylene glycol graft copolymer with amphiphilic properties (PCL-PVAc-PEG) (Figure 1C), is gaining increasing attention. Nanomicelles prepared with Soluplus^®^ are highly stable against dilution due to its low critical micelle concentration (CMC) value [29,30,31], and may also undergo in situ gelling on the ocular surface. Enhanced penetration into ocular structures has been demonstrated for α-lipoic acid [25] and acyclovir [32] encapsulated in Soluplus nanomicelles. The combination of Pluronic and Soluplus has recently been proposed for the oral route, since Soluplus may reinforce the physical stability and drug-loading capacity of Pluronics bearing short PPO blocks [33,34].

The combination of amphiphilic copolymers and cyclodextrins has also been shown to prolong drug permeation at the application site [28,35]. In particular, α-cyclodextrin (αCD) can thread on some block polymers, for example, PEO, and reinforce the hydrophobic interactions among the components through crystalline-like associations among the threaded CDs [36] notably increasing the viscoelastic properties of the system. Such reinforcement is reversible, and thus, under a small amount of pressure, the interactions between αCD molecules are broken, increasing the fluidity of the system. At rest, the bonds between the αCD molecules are regenerated, restoring the gel structure [37].

This work is based on the hypothesis that Soluplus and Pluronic P103 can solubilize natamycin in their micelles, and can also interact with αCD forming poly(pseudo)rotaxanes that are able to tune the rheological properties of the dispersions (Appendix A). To the best of our knowledge, poly(pseudo)rotaxanes of mixed micelles have not been investigated before. It can be assumed that the different composition and architecture of Soluplus and Pluronic P103 in terms of PEG location may strongly determine the properties of the poly(pseudo)rotaxanes formed by each of them and the mixed micelles. The aim of this study was to ascertain the potential of these nanomicelles and poly(pseudo)rotaxanes as nanocarriers for the ocular delivery of natamycin. To address these statements, sets of dispersions were prepared combining Soluplus and Pluronic P103 with αCD, and they were characterized in terms of particle size, zeta potential, polydispersity index, drug solubility, appearance, rheological properties, in vitro drug diffusion, ocular tolerance, and ex vivo permeation.

## 2. Materials and Methods

### 2.1. Materials

Natamycin (665.73 g/mol) was obtained from AK Scientific (San Francisco, CA, USA). Soluplus^®^ (115,000 g/mol) and Pluronic^®^ P103 (EO_17_PO_60_EO_17_; 4950 g/mol) were provided from BASF (Ludwigshafen, Germany). Sodium chloride, MgCl_2_·6H_2_O, and acetonitrile were provided from Scharlab SL (Barcelona, Spain); NaOH, KH_2_PO_4_, NaHCO_3_, CaCl_2_·2H_2_O and NaH_2_PO_4_·H_2_O were provided from Merck (Darmstadt, Germany); KCl from Prolabo (Fontenay-Sous-Bois, France); phosphate-buffered saline (PBS) was provided from Sigma (St. Louis, MO, USA); penicillin and streptomycin were provided from Gibco (Grand Island, NY, USA), and ethanol absolute was provided from Panreac (Barcelona, Spain). α-Cyclodextrin (αCD) was obtained from Wacker Chemie (Munich, Germany).

Phosphate buffer pH 6.4 was prepared by mixing 250 mL of solution A (0.2 M 62.5 mL KH_2_PO_4_) and solution B (0.2 M 16.4 mL NaOH) with water. Carbonate buffer pH 7.2 was prepared by mixing buffer solution A (100 mL; 1.24 g NaCl, 0.071 g KCl, 0.02 g NaH_2_PO_4_, 0.49 g NaHCO_3_) and buffer solution B (100 mL; 0.023 g CaCl_2_, 0.031 g MgCl_2_).

### 2.2. Micelles Preparation and Characterization

Micelles were formed by dispersing Soluplus and Pluronic P103 in different concentrations (0.1%, 0.01%, 1%, 2%, 3%, 4%, 5% *w*/*v*) in 0.9% NaCl. Also, Soluplus and Pluronic P103 at 10% (*w*/*v*) were prepared in pH 6.4 buffer and 0.9% NaCl aq solution. They were kept under magnetic stirring for 12 h. Micelles containing natamycin (0.4 mg drug/mL dispersion) were prepared to compare with unloaded micelles.

Size, zeta potential, and polydispersion index (PDI) were measured using Zetasizer^®^ 3000HS (Malvern Instruments, Malvern, UK). The pH was recorded using pH meter GLP22 (Crison Instruments, L’Hospitalet de Llobregat, Spain). Micelle stability against dilution was recorded for dispersions of Pluronic P103 containing natamycin, which were poured into quartz cells that contained either 0.9% NaCl or pH 6.4 buffer for a sudden 30-fold or 60-fold dilution. The absorbance was recorded at 304 nm every 30 s for 30 min (UV-Vis spectrophotometer Agilent 8453, Waldbronn, Germany). All experiments were carried out in triplicate.

### 2.3. Solubility of Natamycin in Micelle Dispersions

Soluplus and Pluronic P103 dispersions were prepared by adding the required amount of polymer (0.01%, 0.1%, 1%, 2%, 3%, 4%, 5%, and 10% *w*/*v*) in 0.9% NaCl under stirring. Mixtures were obtained by mixing Soluplus (10% *w*/*v*) and Pluronic P103 (10% *w*/*v*) dispersions at various volume ratios (1:4, 2:3, 3:2, and 4:1). Aliquots (5 mL) of each dispersion were placed in test tubes, and natamycin was added in excess (~2 mg). All the dispersions were prepared in triplicate and kept under constant agitation (Unitronic, JP Selecta, Barcelona, Spain) for 6 days at 37 ºC. After that, they were centrifuged (centrifuge model 5804R, Eppendorf AG, Germany) at 5000× *g* rpm for 30 min, and supernatants were diluted in ethanol/water (20:80% *v*/*v*) mixture. Natamycin content was determined by UV-Vis spectrophotometer (Agilent 8453, Waldbronn, Germany) at 304 nm using a previously validated method with standard solutions ranging from 2.0 to 20.0 µg/mL. Also, the pH was measured.

In addition, data from the solubility study were used to calculate the following parameters (25, 32):

(a) Molar solubilization capacity (moles of drug that can be solubilized per mol of copolymer forming micelles):(1)$$X=\frac{S_{\mathrm{tot}}{-S}_{w}}{C_{\mathrm{copol}}-\mathrm{CMC}}$$

(b) Micelle–water partition coefficient (ratio between the drug concentration in the micelle and the aqueous phase):(2)$$P=\frac{S_{\mathrm{tot}}{-S}_{w}}{S_{w}}$$

(c) Molar micelle–water partition coefficient that eliminates the *P* dependence on the copolymer concentration, assigning a default concentration of 1 M:(3)$$\mathrm{PM}=\frac{X\cdot (1-\mathrm{CMC})}{S_{w}}$$

(d) Gibbs standard-free energy of solubilization, which was estimated from the molar micelle/water partition coefficient (*PM*):(4)ΔGs=−RT·ln(PM)

(e) The proportion of drug molecules encapsulated in the micelles:(5)$$m_{f}=\frac{S_{\mathrm{tot}}{-S}_{w}}{S_{\mathrm{tot}}}$$

In these equations, *S_tot_* represents the total solubility of natamycin in the micellar solution, *S_w_* is the natamycin solubility in water, *C_copo_*_l_ is the copolymer concentration in each micelle solution, *CMC* is the critical micelle concentration, and *R* is the universal constant of gases.

### 2.4. Solubility of Natamycin in αCD

Solutions of αCD (5% and 10% *w*/*v*) in pH 6.4 buffer or 0.9% NaCl were prepared. Natamycin was added in excess (approximately 8 mg in 10 mL), and they were kept under magnetic stirring at room temperature for 5 days. After that, they were centrifuged (as above) at 5000× *g* rpm for 30 min, and supernatants were diluted in ethanol/water (20:80% *v*/*v*) mixture. Natamycin content was determined by UV-Vis spectrophotometry at 304 nm. The apparent solubility of natamycin in pH 6.4 buffer and 0.9% NaCl without CDs was similarly measured.

### 2.5. Preparation of Poly(pseudo)rotaxanes

Polypseudorotaxanes were prepared mixing solution A (copolymers) and solution B (αCD) in pH 6.4 buffer and 0.9% NaCl media. For solution A, 20% (*w*/*w*) Soluplus or Pluronic P103 were prepared in each media. Once dissolved, natamycin (up to 240 µg/mL) was added to each copolymer dispersion, and the systems were kept under magnetic stirring at room temperature for 24 h. For solution B, 20% (*w*/*v*) αCD was prepared. Solutions A and B were mixed, and the final concentration was 10% (*w*/*w*) copolymers, 10% (*w*/*w*) αCD, and 120 µg/mL natamycin in each medium. Dispersions containing only the copolymers and natamycin at the same final concentration were also prepared for comparison. Changes in turbidity were examined by visual inspection.

### 2.6. Rheological Characterization

The influence of temperature on the storage (G′) and loss (G″) moduli of Soluplus and Pluronic P103 dispersions, with and without αCD, in pH 6.4 buffer or 0.9% NaCl were recorded in a Rheolyst AR-1000 N rheometer (TA Instruments, UK) equipped with an AR2500 data analyzer, a Peltier plate, and cone geometry (6 cm diameter, 2.1°). Studies were performed at a fixed angular frequency of 5 rad/s and an oscillation stress of 0.1 Pa from 20 to 40 °C with a ramp of 2 °C/min.

### 2.7. Diffusion Assays

Natamycin diffusion tests from Soluplus and Pluronic dispersions and poly(pseudo)rotaxanes were performed in triplicate in vertical Franz diffusion cells fitted with cellulose acetate membrane filters (0.45-mm pore size, 25-mm diameter). Aliquots of 1.00 mL of the test formulation at 37 °C were placed in the donor compartment. The receptor phase contained 6.00 mL of medium (0.9% NaCl or pH 6.4 buffer) thermostated at 37 °C and kept under magnetic stirring. The area available for diffusion was 0.786 cm^2^. Samples (0.70 mL) were taken from the receptor phase at 30, 60, 90, 120, 180, 210, 240, 300 and 360 min, and replaced immediately with fresh medium. Natamycin concentration was determined by UV-Vis spectrophotometry.

Diffusion coefficients were estimated by following the Higuchi equation:(6)$$\frac{Q}{A}{=2C}_{0}{\left(\frac{\mathrm{Dt}}{\pi }\right)}^{\frac{1}{2}}$$
where *Q* is the amount of natamycin (g) released by time *t* (min), *A* is the diffusion area (cm^2^), *C*_0_ is the initial concentration of natamycin in the formulation (g/mL), and *D* is the diffusion coefficient (cm^2^/min). The average size and zeta potential of the formulation tested in this assay were also characterized.

### 2.8. Ocular Tolerance Test (HET-CAM Test)

Fertile chicken eggs were kindly donated by The Coren Technological Incubation Center (San Cibrao das Viñas, Spain) and incubated at 37 °C and 60% RH. Eggs were manually rotated 180° three times per day to ensure the correct development of the embryo. After 9 days of incubation, a circular cut (about 1 cm in diameter) was made on the eggshell using a rotatory saw. The inner membrane was wet with 0.9% NaCl, and then carefully removed to expose the chorioallantoic membrane (CAM). Any defective egg was discarded. Aliquots of natamycin-loaded Soluplus and Pluronic P103 micelles (200 μL) and poly(pseudo)rotaxanes (150 μL) dispersions were placed on the CAM of different eggs. Negative and positive controls were 0.9% NaCl and 0.1 N NaOH solutions, respectively. The development of hemorrhage (*tH*, bleeding from the vessels), vascular lysis (*tL*, blood vessel disintegration), or coagulation (*tC*, intravascular and extravascular protein denaturation) of CAM vessels was monitored for 300 s. Then, the irritation score (*IS*) was calculated as follows [25]:(7)$$\mathrm{IS}=\frac{(301-\mathrm{tH})\times 5}{300}+\frac{\left(301-\mathrm{tL}\right)\times 7}{300}+\frac{\left(301-\mathrm{tC}\right)\times 9}{300}$$

The damage was classified by means of *IS* as non-irritant (*IS* < 1), mild irritant (1 ≤ *IS* < 5), moderately irritant (5 ≤ *IS* < 10), or severe irritant (*IS* > 10). Each test was performed at least in duplicate.

### 2.9. Ex-Vivo Corneal and Sclera Permeability Study

Bovine eyes were collected immediately after sacrifice from a local slaughterhouse and transported immersed in PBS solution containing antibiotics (penicillin 100 IU/mL and streptomycin 100 μg/mL) and maintained in an ice bath. Next, corneas and scleras were isolated, rinsed with PBS, and placed on vertical diffusion Franz cells between the donor and receptor compartments. Both compartments were filled with carbonate buffer pH 7.2. The receptors were kept immersed inside a bath at 37 °C, and gentle magnetic stirring was applied for 1 h in order to balance ocular tissues. Then, the buffer at the donor chamber was completely removed and replaced by the formulations (2 mL) prepared as described above either in pH 6.4 buffer or 0.9% NaCl. The donor compartments were covered with parafilm (0.785 cm^2^ area available for permeation). Samples (1 mL) were taken from the receptor compartment at 0.5, 1, 2, 3, 4, 5 and 6 h, and the same volume was replaced with fresh medium, taking care to remove bubbles from the diffusion cells. All the experiments were carried out in triplicate.

Natamycin permeated was quantified using a Jasco (Tokyo, Japan) HPLC (AS-4140 Autosampler, PU-4180 Pump, LC-NetII/ADC Interface Box, CO-4060 Column Oven, MD-4010 Photodiode Array Detector), fitted with a C18 column (Waters Symmetry C18, 5 μm, 3.9 × 150 mm) and operated using ChromNAV software (ver. 2, Jasco, Tokyo, Japan). The mobile phase was acetonitrile: 30 mM of perchlorid acid (35:65) at 1 mL/min and 30 °C. The injection volume was 90 µL, and natamycin was quantified at 304 nm (retention time 3.3 min). Standard solutions of natamycin (0.01–1 µg/mL) in ethanol/water (20:80) were prepared.

After a 6-h permeation test, drug concentration in the donors was quantified. All the corneas and scleras were visually inspected after the 6-h test to verify that none of them had cracks or modified their appearance. Corneas and scleras were soaked in ethanol:water (50:50 *v*/*v*; 3 mL) medium at 37 °C during 24 h. Then, they were sonicated during 99 min at 37 °C, centrifuged (1000 rpm, 5 min, 25 °C), and the supernant was filtered (Acrodisc^®^ Syringe Filter, 0.22 µm GHP Minispike, Waters), centrifuged again (14,000× *g* rpm, 20 min, 25 °C), and filtered to be measured in HPLC.

The apparent permeability coefficient (*P_app_*) was calculated from the flux (*J*) according to Equation (8) [32]:(8)$$P_{\mathrm{app}}=\frac{J}{C_{0}}$$
where *J* is the flux calculated as the slope (*Q*/*t*) of the linear section of the amount of drug in the receptor chamber (*Q*) versus time (*t*), and *C*_0_ is the initial concentration of natamcyin in the donor phase. Each experiment was performed in triplicate, and the results were reported as the mean values ± standard deviation (SD).

### 2.10. Statistical Analysis

The effects of formulation composition on natamycin permeation through sclera were analyzed using ANOVA and a multiple range test (Statgraphics Centurion XVI 1.15, StatPoint Technologies Inc., Warrenton VA, USA).

## 3. Results and Discussion

### 3.1. Micelles Preparation and Natamycin Solubilization

The first experiments carried out in 0.9% NaCl aqueous medium revealed that Soluplus micelles were larger (70–90 nm) and conferred more acidic environment (~pH 4) than those of Pluronic P103 (20 nm; ~pH 6) (Table 1 and Appendix A). The zeta potential was slightly negative in all the cases. The reported CMCs for Soluplus and Pluronic P103 are 0.00076% *w*/*v* (i.e., 6.60 × 10^−8^ M) [25,38] and 0.070% *w*/*v* (1.41 × 10^−4^ M) [39], which is in good agreement with our results.

Mixtures of 10% (*w*/*v*) Soluplus and 10% (*w*/*v*) Pluronic dispersions prepared at various volume ratios (1:4, 2:3, 3:2, and 4:1) in 0.9% NaCl led to intermediate pH values and significantly larger micelles (Table 1). Studies performed by Koontz et al. [18] revealed that natamycin is more stable at pH ranging from 4 to 7. Thus, for the sake of comparison of the micelle properties avoiding changes in pH, the micelles and their mixtures were also prepared in pH 6.4 buffer (Table 1). Once again, a remarkable increase in the micelle size was observed as the Pluronic P103 proportion increased in the mixed micelles, which suggests that the large PPO block of Pluronic P103 (EO_17_PO_60_EO_17_) accommodates inside the Soluplus cores. Commonly, an increase in micelle size is associated with the expansion of the core, and it should be noted that Pluronic P103 is notably more hydrophobic (HLB = 9) [40] than Soluplus (HLB = 16) [32].

The apparent solubility of natamycin in 0.9% NaCl (0% block copolymer) was 43.82 ± 1.76 µg/mL, which agrees well with the values previously reported [11]. As expected, increasing the concentration of copolymer in the medium resulted in increased natamycin apparent solubility, confirming natamycin incorporation into the nanomicelles (Figure 2 and Figure 3).

The increase in apparent solubility was evident for both copolymers at concentrations equals to or above 1%, but the increase was significantly greater for the Soluplus dispersions. The apparent solubility of natamycin reached 138.6 ± 1.9 µg/mL in Soluplus 5% *w*/*v* in 0.9% NaCl at 25 °C. The high solubilizing capability of Soluplus has been associated to its large core, which in turn leads to greater micelles [38]. Parameters related to micelle solubilizing efficiency are summarized in Table 2 and Table 3. The molar solubilization capability, χ, was nearly two orders of magnitude greater for Soluplus compared to Pluronic P103. The χ value decreased with increasing copolymer concentration, which indicates that more unimers are involved in micelle formation. The encapsulation of natamycin occurred spontaneously for both micelle types, but the thermodynamics (i.e., standard-free Gibbs energy of solubilization) were more favorable for Soluplus systems. For a fixed concentration of 5% *w*/*v* copolymer, ca. 70% drug molecules were encapsulated into the Soluplus micelles, while only 50% were encapsulated in the case of Pluronic P103. The stability of the micelles against dilution could be only verified for the Pluronic nanomicelles (Appendix A), since the Soluplus nanomicelles absorbed light in the same UV region as the drug. This low Soluplus UV absorption did not interfere with drug solubility quantification in the apparent solubility studies as the micelles disassemble in the ethanol–water medium, but it caused noise under the dilution with aqueous medium of the stability test. In the case of Pluronic P103, after strong dilution in either 0.9% NaCl solution or pH 6.4 buffer, the absorbance showed a minor initial decrease followed by a complete and stable recovery, which indicates rapid rebalancing of the micelle–medium partition equilibrium.

Soluplus and Pluronic concentrations were increased to 10% *w*/*v*, and their solubilization capability was compared to that of the mixed micellar solutions (1:4, 2:3, 3:2, 4:1 *v*/*v*) prepared in 0.9% NaCl and pH 6.4 buffer (Figure 3). Once again, the Soluplus (10% *w*/*v*) micelles dispersion showed greater apparent solubility enhancement, especially in pH 6.4 buffer (263.7 ± 10.9 µg/mL) compared to 0.9% NaCl medium (198.5 ± 2.3 µg/mL). The apparent solubility of natamycin in the Soluplus /Pluronic mixtures ranged from 86.8 ± 0.3 to 123.8 ± 1.3 µg/mL in 0.9% NaCl, and from 89.7 ± 11.9 to 119.1 ± 0.6 µg/mL in pH 6.4 buffer. The higher the content in P103, the lower the solubilization capability. The partition coefficients P for Soluplus/Pluronic P103 1:4 and 4:1 (*v*/*v*) were in the ranges 0.98 to 1.05 and 1.72 to 1.82, respectively. This finding correlates with the increase in size observed for the mixed micelles, which suggests that Pluronic P103 occupies the core of Soluplus micelles and alters the inner hydrophobic interactions, which in turn hinders the accommodation of natamycin. Similar behavior was observed by Bernabeu et al. [38] for mixed systems formed with Soluplus and D-α-tocopheryl polyethylene-glycol 1000 succinate (TPGS). The incorporation of TPGS into Soluplus micelles made the self-assembly of Soluplus molecules more difficult, increased the size of the micelles, and notably decreased their capability to host paclitaxel. These detrimental effects attenuated as the Soluplus ratio increased. It should be noted that compared to previous reports on Soluplus/Pluronic mixed micelles, the variety that we tested (Pluronic P103) is more hydrophobic than those evaluated before (P105 and F127) [33,34], which may explain the stronger effects on the properties of the mixed micelles. For subsequent studies, only the mixed micelles with the largest ratio in Soluplus (i.e., 4:1), and thus with the size more similar to pure Soluplus micelles were considered.

### 3.2. Poly(pseudo)rotaxane Formation

Previous to the preparation of the poly(pseudo)rotaxanes, the capability of αCD to form an inclusion complex with natamycin, which is more soluble than free natamycin, was verified. The information available on the use of αCD to solubilize ocular antifungal drugs is so far very limited [41]. The apparent solubility of natamycin in aqueous 5% and 10% (*w*/*v*) αCD solutions was determined to be 103.7 ± 1.4 µg/mL and 143.5 ± 4.6 µg/mL in 0.9% NaCl medium, and 96.4 ± 2.9 µg/mL and 126.6 ± 0.4 µg/mL in pH 6.4 buffer, respectively (Appendix A). These values indicate an efficient encapsulation of the drug in αCD cavities, achieving apparent solubility values similar to those obtained for Pluronic P103 micellar dispersions at the same concentrations, although lower than those provided by Soluplus. A further increase in αCD concentration was not tested due to safety concerns [42].

Based on these preliminary data, 10% Soluplus, 10% Pluronic P103, and binary systems containing Soluplus 10%/Pluronic P103 10% (4:1 *v*/*v*) in aqueous pH 6.4 buffer were selected for poly(pseudo)rotaxane formation. Each dispersion was prepared at double concentration and then mixed in vortex for a few minutes. Natamycin concentration was set at 120 µg/mL, which is three-fold greater than the apparent drug solubility in water, but still below the solubilizing capability of each separate component (copolymer/αCD). The addition of αCD solution up to a final concentration of 10% caused remarkable changes in gel appearance and rheological properties. The initial transparent (Pluronic P103) or opalescent (Soluplus) dispersions rapidly transformed into whitish systems, which indicated poly(pseudo)rotaxane formation. All these changes occurred both in 0.9% NaCl or pH 6.4 buffer (Figure 4).

Poly(pseudo)rotaxane formation was also confirmed by recording the viscoelastic behavior of the copolymer dispersions with and without αCD (Figure 5). Pluronic P103 10% *w*/*v* dispersions were performed as a liquid-like material with negligible storage modulus (G′). Moreover, G″ values remained constant in the temperature range evaluated; namely, a sol-to-gel transition was not observed. The addition of αCD caused a remarkable increase in both G′ and G″; the G′ values became larger than the G″ ones. The poly(pseudo)rotaxane system behaved as weak gels in the 20 to 40 °C range. αCD has been reported to be able to rapidly thread along PEO blocks, decreasing their hydrophilicity and favoring the stacking of the PEO blocks through channel-type interactions between the threaded αCDs [37]. This stacking reinforces the tie junctions among the poloxamer molecules, which in the absence of αCDs are only driven by the PPO–PPO hydrophobic interactions. The presence of natamycin caused minor changes in the viscoelastic performance of the Pluronic P103-based poly(pseudo)rotaxanes.

Differently, Soluplus 10% *w*/*v* dispersions exhibited a peculiar dependence on temperature. In agreement with previous reports (25, 32), G′ values were above 1 Pa at approximately 34 °C, and then rose in parallel with those of G″. Thus, instead of showing a sudden sol-to-gel transition, the values of both G′ and G″ progressively increased with the temperature. This behavior indicates a poorly cooperative hydrophobic-driven transition, which was probably due to the particular configuration of the Soluplus copolymer with three blocks of different hydrophilicity. The addition of αCD to unloaded Soluplus dispersion caused a remarkable increase in both moduli. As in the case of Pluronic P103, the Soluplus-based poly(pseudo)rotaxanes performed as weak gels already at 20 °C, but showed a further increase in G′ and G″ values at temperatures above 30 °C. The drug-loaded dispersions behaved quite differently. They maintained low G′ and G″ values at room temperature and exhibited a sol-to-gel transition at 30.5 °C, which is a transition temperature that is significantly lower than that recorded in the absence of αCD. Differences between unloaded and natamycin-loaded Soluplus dispersions suggest that the hosting of natamycin in the Soluplus micelles alter the copolymer capability to form strong complexes with αCD at room temperature. Nevertheless, at the temperature of the eye surface, the values of G′ and G″ are at least one order of magnitude larger for the poly(pseudo)rotaxanes compared to the Soluplus micelle dispersions.

To the best of our knowledge, the poly(pseudo)rotaxane formation of mixed micelles has not been previously investigated. Soluplus/P103 4:1 dispersions showed a shift in the sol-to-gel transition toward higher values (36.1 °C) compared to Soluplus solely dispersion, except for the unloaded system in NaCl 0.9%. This finding agrees well with the distortion of the self-assembly process of Soluplus as Pluronic P103 accommodates in the micelle core (as hypothesized from the increase in micelle size). The addition of αCD to the mixed micelles remarkably increased the viscoelastic parameters. The system performed as a weak gel already at 20 °C, and only a slight increase in G′ and G″ was observed during heating. Interestingly, the presence of natamycin only caused a minor decrease in both G′ and G″, which means that Pluronic P103 attenuated the effects of hosting natamycin in Soluplus cores probably because of a preferential interaction of αCD molecules with the PEO blocks of Pluronic P103. It has been previously observed by means of nuclear magnetic resonance diffusion studies that the threading of αCD along PEO moieties of Pluronic F127 causes a remarkable decrease in αCD mobility [43], which is typical of strong complex formation. The decrease in the mobility of αCDs due to the formation of a transient complex with Soluplus was less marked [35], which supports our hypothesis of a preferential threading of αCDs along Pluronic P103 in the mixed micelles.

### 3.3. HET-CAM Assay

The preliminary screening of potential irritancy of the formulations was performed using the Hen’s Egg Test on the Chorioallantoic Membrane (HET-CAM), which is an alternative to Draize rabbit eye test [25,44,45]. The HET-CAM test revealed that no micelle or poly(pseudo)rotaxane dispersions caused hemorrhage, lysis, or coagulation at the time of study (*tH, tL*, and *tC* values ≫ 301 s) (Appendix A). The *IS* was 0.0 for all the formulations, as occurred with the negative control (0.9% NaCl). Therefore, they can be considered as non-irritants. Differently, the *IS* for the positive control (NaOH 0.1N) was 18.58. This behavior is consistent with previously reported studies [25,35].

### 3.4. Natamycin Diffusion

Natamycin diffusion from micelles, mixed micelles, and poly(pseudo)rotaxanes prepared in 0.9% NaCl and pH 6.4 buffer was first evaluated in vitro (under sink conditions) using a membrane with a large (0.45 mm) pore size in order to quantify the capability of the formulations to control drug release. All the dispersions showed sustained diffusion (Figure 6), but differences were observed depending on the copolymer and the addition of αCD. In the case of micelles, the Pluronic P103 10% *w*/*v* formulation provided the fastest diffusion (46.45 ± 1.41 µg/cm^2^ in 0.9% NaCl and 48.25 ± 0.42 µg/cm^2^ in buffer pH 6.4 after 6 h), followed by mixed micelles (25.69 ± 0.50 µg/cm^2^ in 0.9% NaCl and 42.25 ± 1.72 µg/cm^2^ in buffer pH 6.4 after 6 h) and Soluplus micelles (13.39 ± 1.04 µg/cm^2^ in 0.9% NaCl and 20.13 ± 0.95 µg/cm^2^ in buffer pH 6.4 after 6 h). This finding is in agreement with the smaller size of Pluronic P103 micelles and the less compact structure of these micelles and also of the mixed micelles compared to those of Soluplus. The partition coefficient, P, for Soluplus/Pluronic P103 4:1 (*v*/*v*) was in the 1.72 to 1.82 range. These values are lower than those recorded for pure Soluplus or Pluronic in separate. This means that there were more free drug molecules in the mixed micelles. Nevertheless, the diffusion was slower than that from Pluronic micelles, which supported the hypothesis that the smaller size of Pluronic micelles plays a relevant role in the diffusion rate, although other factors such as copolymer composition and drug–core interactions may also affect the diffusion kinetics.

Regarding poly(pseudo)rotaxanes, the highest amount of natamycin that had diffused after 6 h corresponded to Pluronic P103 formulation (33.76 ± 2.86 µg/cm^2^ in 0.9% NaCl and 45.76 ± 4.79 µg/cm^2^ in pH 6.4 buffer), followed by poly(pseudo)rotaxanes of mixed micelles (17.71 ± 1.08 µg/cm^2^ in 0.9% NaCl and 23.60 ± 2.42 µg/cm^2^ in pH 6.4 buffer) and Soluplus-based poly(pseudo)rotaxanes (12.99 ± 0.19 µg/cm^2^ in 0.9% NaCl and 18.78 ± 1.55 µg/cm^2^ in pH 6.4 buffer). Soluplus poly(pseudo)rotaxanes showed the slowest diffusion rate. The delay in drug diffusion observed for poly(pseudo)rotaxanes is related to the formation of a more structured network with higher viscosity [35], as recorded in the rheometry study. Nevertheless, the increase in macroviscosity might not directly translate to higher microviscosity [46]. Indeed, in the case of Soluplus, it has been previously shown for carvedilol transdermal formulations that the differences in drug release between micelles and poly(pseudo)rotaxanes are minor [35]. The diffusion coefficients (D) of natamycin calculated applying Equation (6) are shown in Table 4.

Correlation coefficients (R^2^) confirmed that the Higuchi´s kinetics fitted well. Small differences in drug diffusion coefficients were obtained when the formulations were prepared and tested in pH 6.4 buffer compared to 0.9% NaCl. Diffusion coefficients were almost one order of magnitude lower for Soluplus micelles compared to Pluronic micelles, and for both formulations, the addition of αCD caused a small decrease in the diffusion coefficients. The poly(pseudo)rotaxanes made from mixed micelles showed natamycin diffusion coefficients intermediate between those of Pluronic P103-based and Soluplus-based poly(pseudo)rotaxanes. Therefore, the preparation of mixed micelles may be a useful tool to regulate drug release from poly(pseudo)rotaxanes.

### 3.5. Ex Vivo Permeation Assay

Bovine cornea and sclera were isolated from fresh eyes and mounted in vertical (Franz) diffusion cells using carbonate buffer pH 7.2 as the receptor medium (37 °C). Natamycin that permeated through ocular tissues to the receptor chamber was monitored for 6 h. In the case of cornea permeation tests, the levels of drug in the receptor compartment were below the quantification limit (0.01 µg/mL) during the first five hours. At 6 h, only the micelle formulations gave measurable natamycin amounts, and the amounts permeated ranged between 0.13–0.26 µg/cm^2^, disregarding the copolymer involved. In spite of this low ability to cross the cornea, natamycin accumulated in the cornea tissue at the relevant amount (Figure 7A,B). Interestingly, for each copolymer tested, the poly(pseudo)rotaxanes provided higher amounts of drug accumulated than the corresponding micelles. Soluplus-based poly(pseudo)rotaxanes displayed the highest corneal accumulation, disregarding whether the formulations were prepared in 0.9% NaCl or pH 6.4 buffer (4.24 ± 0.13 µg/cm^2^ and 7.50 ± 0.03 µg/cm^2^, respectively).

Compared to the in vitro tests in which the membrane is not an effective barrier to drug diffusion, the compact layered structure of the cornea is a quite challenging barrier for any drug, and some studies have shown important diffusion lag times due to the difficult movement of the drugs into the cornea. Indeed, transcorneal penetration studies carried out using ex vivo rabbits´ eyes with Natacyn^®^ (suspension diluted to 3 mg drug per mL) revealed quite low transcorneal flux (0.14 ± 0.1 × 10^−6^ cm/s) [15]. Our formulations had all the same initial concentration in natamycin (120 µg/mL), which was ca. 3 times larger than the apparent solubility in water and above the highest minimal inhibitory concentration (MIC_90_ = 64 µg/mL) reported for clinically relevant fungi species [47]. The increase in apparent drug solubility once encapsulated in the micelles and poly(pseudo)rotaxanes should lead to an increase in drug gradient concentration on the tissue surface, which should favor drug penetration. Nevertheless, the larger size of encapsulating species and restrictions imposed by the tissue itself may hinder passive drug diffusion. In this sense, the presence of αCD in the polypseudorotaxanes may facilitate the mobility of natamycin in the aqueous layer in contact with the cornea, and may also facilitate the entrance in the first hydrophilic layer [48,49], disregarding the apparent macroviscosity of the formulations. To cross the cornea, the drug is expected to abandon the micelles/CDs. Nevertheless, free drug permeation through the cornea depends on the drug lipophilicity and, in the particular case of natamycin, the LogP is about 1.1 [50], which seems to be quite low for facilitating the transcorneal penetration.

Sclera usually exhibits larger permeability for most drugs compared to the cornea due to its larger pores, which even allow the penetration of nanoparticles with sizes in between 20–200 nm [51]. Although in vivo, the surface area of absorption in the sclera is larger than that in the cornea [52], for comparative purposes in the ex vivo studies, the area available for diffusion was the same as that for the cornea tests. Larger amounts of natamycin accumulated in the sclera (Figure 7C,D), and once again, the poly(pseudo)rotaxanes seemed to facilitate the accumulation. Differently to the cornea, natamycin was able to permeate through sclera toward the receptor chamber. After a lag time of approximately 1 h, natamycin concentration steadily increased in the receptor medium (Figure 8). The amounts permeated were larger for the micelles than for the poly(pseudo)rotaxanes, and ordered inversely to the size of the carrier. Namely, the small Pluronic P103 micelles permeated faster than the larger Soluplus micelles and the mixed micelles.

The amount of natamycin permeated per surface area showed a linear dependence on time. The permeability coefficient (*P_app_*) of natamycin across bovine sclera was calculated as the ratio of flux (*J*) and the concentration of natamycin in the donor phase (Table 5). In all the cases, the poly(pseudo)rotaxanes led to a smaller apparent permeability coefficient than the corresponding micelles (ANOVA and multiple range test; F_5,12df_ = 90.73; *p* < 0.001 in 0.9% NaCl; F_5,12df_ = 377.62; *p* < 0.001 in buffer pH 6.4). The smallest permeability coefficient was for Soluplus poly(pseudo)rotaxane, while the highest *P_app_* corresponded to Pluronic P103 solely micelles.

## 4. Conclusions

Soluplus and Pluronic P103 micelles as well as αCD can encapsulate natamycin, although the capability of Soluplus micelles to increase the drug apparent solubility is significantly greater. To the best of our knowledge, the formation of poly(pseudo)rotaxanes using mixed micelles has not been previously reported. Interestingly, mixed micelles of Soluplus and Pluronic show a remarkable increase in the micelle size, which suggests that the large PPO block of Pluronic P103 accommodates inside the Soluplus cores. As a consequence, the drug solubilization capability of mixed micelles is lower than that of Soluplus micelles alone. The addition of αCD to the dispersions of micelles caused a remarkable increase in both G′ and G″, and this increase was reinforced by the temperature-responsiveness of the copolymers. The increase in the apparent macroviscosity of the poly(pseudo)rotaxanes on the ocular surface conditions may be useful for prolonging their permanence. Relevantly, the increase in the macroviscosity of the poly(pseudo)rotaxanes causes a decrease in drug diffusion, especially in the case of the mixed micelles system. Overall, mixed micelles and their poly(pseudo)rotaxanes allow for tuning the features that each copolymer system exhibits separately, i.e., mixing Soluplus and Pluronic leads to intermediate natamycin diffusion, cornea and sclera accumulation, and sclera permeability coefficients, without detrimental effects on biocompatibility. Therefore, the poly(pseudo)rotaxanes of mixed micelles are identified as technological tools that are suitable for the controlled release of poorly soluble drugs.

## Figures and Tables

**Figure 1 nanomaterials-09-00745-f001:**
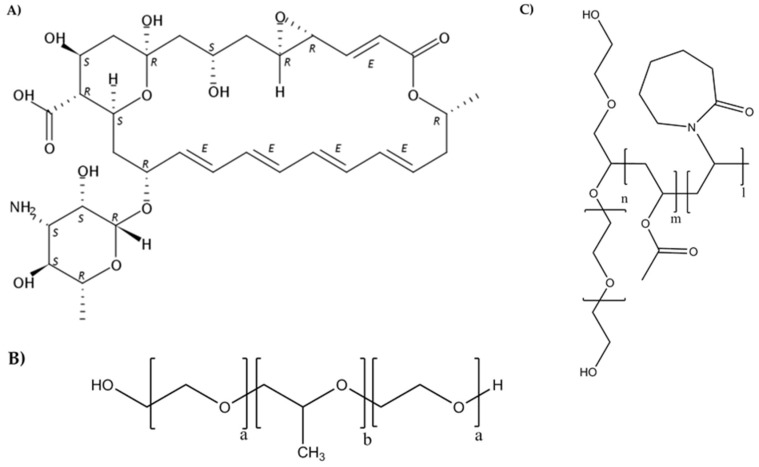
Chemical structures of natamycin (**A**), Pluronic^®^ (**B**), and Soluplus^®^ (**C**).

**Figure 2 nanomaterials-09-00745-f002:**
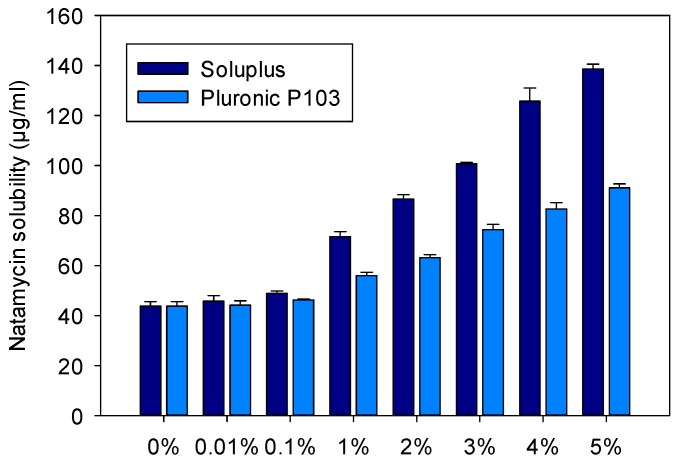
Apparent solubility of natamycin in Soluplus and Pluronic P 103 dispersions (0%, 0.01%, 0.1%, 1%, 2%, 3%, 4%, and 5% *w*/*v*) prepared in 0.9% NaCl at 25 °C.

**Figure 3 nanomaterials-09-00745-f003:**
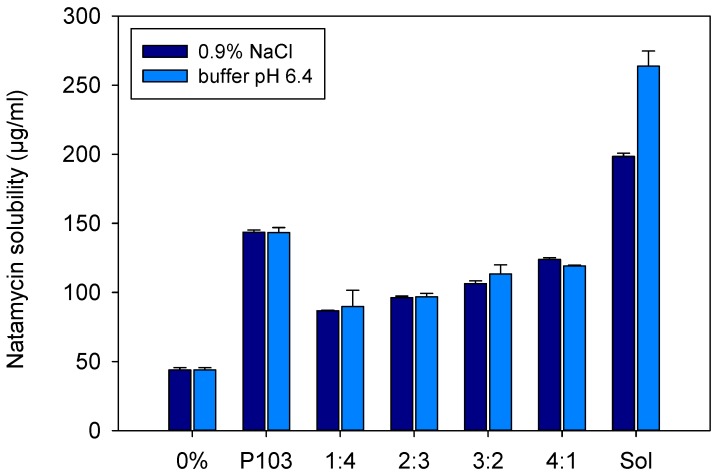
Apparent solubility of natamycin in micelle dispersions of 10% (*w*/*v*) of Soluplus and Pluronic P103 and their mixtures (Soluplus:Pluronic P103) prepared at various volume ratios in 0.9% NaCl and pH 6.4 buffer. Total copolymer concentration was 10% *w*/*v* in all cases.

**Figure 4 nanomaterials-09-00745-f004:**
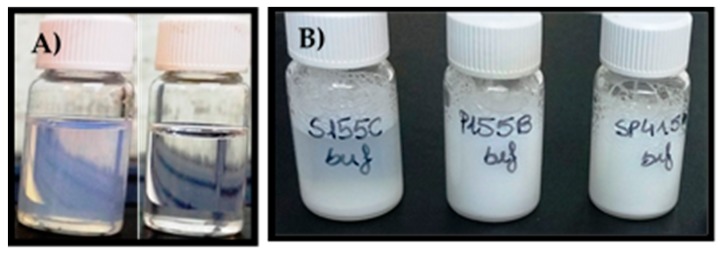
Appearance of (**A**) Soluplus 10% (left) and Pluronic P103 10% (right) dispersions in 0.9% NaCl; (**B**) Poly(pseudo)rotaxane formation after the addition of 10% αCD to Soluplus (left), Soluplus/Pluronic P103 in ratio (4:1) (in the middle), and Pluronic P103 dispersions (right) in pH 6.4 buffer after storage for 12 h at room temperature.

**Figure 5 nanomaterials-09-00745-f005:**
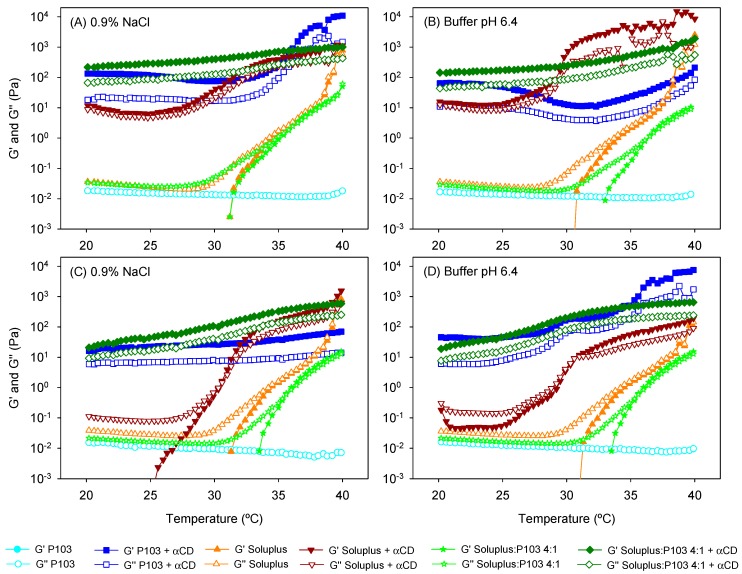
Evolution of the storage (G′) and the loss (G″) moduli as a function of temperature of (**A**,**B**) unloaded and (**C**,**D**) drug-loaded copolymer dispersions and their mixtures with and without α-cyclodextrin (αCD) (poly(pseudo)rotaxanes) in 0.9% NaCl (left) and pH 6.4 buffer (right). Total copolymer concentration was 10% *w*/*v* in all cases.

**Figure 6 nanomaterials-09-00745-f006:**
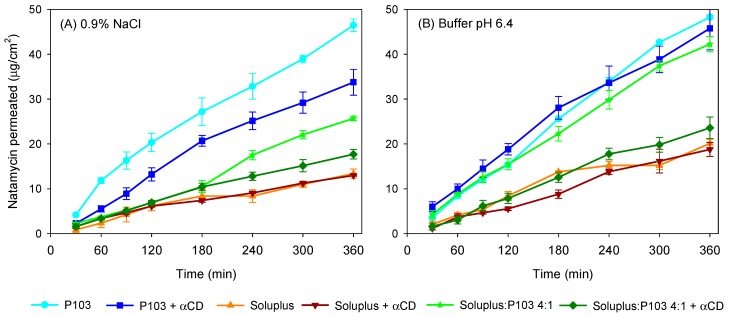
Natamycin diffusion test at 37 °C from Soluplus and Pluronic micelles, Soluplus:Pluronic P103 4:1 *v*/*v* mixed micelles, and poly(pseudo)rotaxanes in 0.9% NaCl (**A**) and pH 6.4 buffer (**B**). Total copolymer concentration was 10% *w*/*v* in all cases.

**Figure 7 nanomaterials-09-00745-f007:**
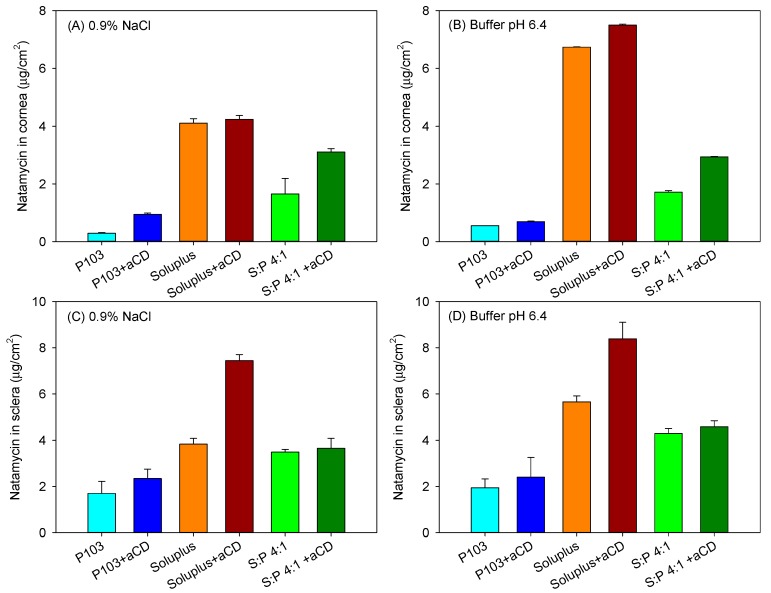
Amount of natamycin accumulated inside bovine cornea (**A**,**B**) and sclera (**C**,**D**) after 6 h in contact with Soluplus and Pluronic micelles, Soluplus:Pluronic P103 4:1 *v*/*v* mixed micelles, and their poly(pseudo)rotaxanes in 0.9% NaCl (left) and pH 6.4 buffer (right). Total copolymer concentration was 10% *w*/*v* in all cases.

**Figure 8 nanomaterials-09-00745-f008:**
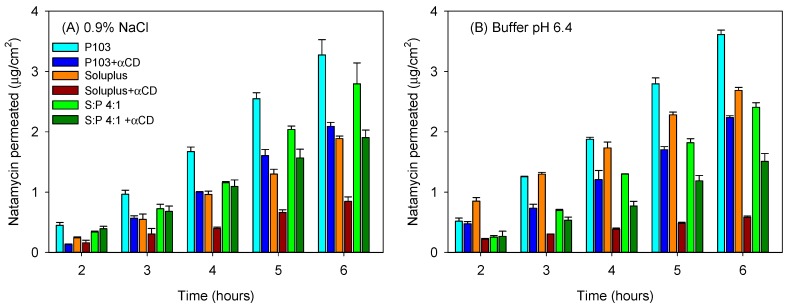
Amount of natamycin permeated through bovine sclera and measured in the receptor chamber as a function of time. Natamycin was formulated in Soluplus and Pluronic micelles, Soluplus:Pluronic P103 4:1 *v*/*v* mixed micelles, and their poly(pseudo)rotaxanes in 0.9% NaCl (**A**) and pH 6.4 buffer (**B**). Total copolymer concentration was 10% *w*/*v* in all cases.

**Table 1 nanomaterials-09-00745-t001:** The pH, size, and zeta potential of unloaded micelles of Soluplus and Pluronic P103 and their mixtures prepared at various volume ratios in 0.9% NaCl and pH 6.4 buffer.

**0.9% NaCl**				
**Copolymer (%*w*/*v*)**	**pH**	**Diameter (nm)**	**PDI**	**Zeta Potential (mV)**
Soluplus (10%)	3.34	90.0 ± 1.3	0.168 ± 0.010	−0.40 ± 0.19
Pluronic (10%)	6.34	20.5 ± 0.6	0.238 ± 0.003	1.03 ± 0.47
Soluplus/Pluronic P103 (1:4)	4.67	129.6 ± 2.9	0.246 ± 0.019	−0.66 ± 0.20
Soluplus/Pluronic P103 (2:3)	3.89	131.0 ± 3.0	0.214 ± 0.011	−0.81 ± 0.08
Soluplus/Pluronic P103 (3:2)	3.70	121.7 ± 1.0	0.190 ± 0.017	−1.49 ± 0.46
Soluplus/Pluronic P103 (4:1)	3.52	110.7 ± 1.8	0.209 ± 0.008	−0.56 ± 0.41
**Buffer pH 6.4**				
**Copolymer (%*w*/*v*)**	**pH**	**Diameter (nm)**	**PDI**	**Zeta Potential (mV)**
Soluplus (10%)	6.08	102.8 ± 1.0	0.189 ± 0.018	−0.14 ± 0.36
Pluronic (10%)	6.36	16.1 ± 0.4	0.226 ± 0.018	1.15 ± 0.28
Soluplus/Pluronic P103 (1:4)	6.49	150.8 ± 4.5	0.217 ± 0.011	0.48 ± 0.06
Soluplus/Pluronic P103 (2:3)	6.47	140.5 ± 0.7	0.176 ± 0.008	−0.02 ± 0.15
Soluplus/Pluronic P103 (3:2)	6.20	127.8 ± 0.9	0.171 ± 0.011	−0.12 ± 0.01
Soluplus/Pluronic P103 (4:1)	6.48	114.7 ± 2.4	0.170 ± 0.011	−0.18 ± 0.13

**Table 2 nanomaterials-09-00745-t002:** Capability of Soluplus dispersions in 0.9% NaCl to solubilize natamycin, estimated using Equations (1)–(5). (NAT: natamycin; χ: molar solubilization capacity; P: partition coefficient; PM: molar partition coefficient; ΔG: standard-free Gibbs energy of solubilization; mf: molar fraction of drug encapsulated inside the micelle). * Data from solubility experiments carried out in pH 6.4 buffer.

Copolymer (% *w*/*w*)	Soluplus (M)	NAT (M)	NAT (µg/mL)	χ	P	PM	ΔG (KJ/mol)	mf
0.1	0.87 × 10^−5^	0.71 × 10^−4^	46.99	0.52	0.07	7869.6	−22,227.0	0.06
1	8.70 × 10^−5^	1.04 × 10^−4^	69.31	0.44	0.58	6619.2	−21,798.2	0.37
2	1.74 × 10^−4^	1.26 × 10^−4^	84.03	0.35	0.91	5233.1	−21,216.1	0.48
3	2.61 × 10^−4^	1.47 × 10^−4^	97.93	0.31	1.23	4700.0	−20,949.8	0.55
4	3.48 × 10^−4^	1.84 × 10^−4^	122.51	0.34	1.78	5131.2	−21,167.3	0.64
5	4.35 × 10^−4^	2.03 × 10^−4^	135.23	0.31	2.07	4769.5	−20,986.2	0.67
10	8.70 × 10^−4^	2.98 × 10^−4^	198.49	0.27	3.51	4038.0	−20,573.7	0.78
10 (buffer) *	8.70 × 10^−4^	3.96 × 10^−4^	263.73	0.38	4.99	5743.2	−21,446.5	0.83

**Table 3 nanomaterials-09-00745-t003:** Capability of Pluronic P103 dispersions in 0.9% NaCl to solubilize natamycin, estimated using Equations (1)–(5). (NAT: natamycin; χ: molar solubilization capacity; P: partition coefficient; PM: molar partition coefficient; ΔG: standard-free Gibbs energy of solubilization; mf: molar fraction of drug encapsulated inside the micelle). * Data from solubility experiments carried out in pH 6.4 buffer.

Copolymer (% *w*/*w*)	Pluronic P103 (M)	NAT (M)	NAT (µg/mL)	χ	P	PM	ΔG (KJ/mol)	mf
0.1	0.20 × 10^−3^	0.70 × 10^−4^	46.27	5.59 × 10^−2^	0.05	845.5	−16,699.6	0.05
1	2.02 × 10^−3^	0.84 × 10^−4^	56.00	9.59 × 10^−3^	0.27	145.1	−12,332.5	0.21
2	4.04 × 10^−3^	0.95 × 10^−4^	63.21	7.40 × 10^−3^	0.44	112.0	−11,690.4	0.30
3	6.06 × 10^−3^	1.12 × 10^−4^	74.37	7.71 × 10^−3^	0.69	116.6	−11,790.1	0.41
4	8.08 × 10^−3^	1.24 × 10^−4^	82.63	7.31 × 10^−3^	0.88	110.6	−11,659.3	0.47
5	1.01 × 10^−2^	1.37 × 10^−4^	91.06	7.10 × 10^−3^	1.07	107.4	−11,586.3	0.52
10	2.02 × 10^−2^	2.16 × 10^−4^	143.49	7.45 × 10^−3^	2.27	112.7	−11,706.4	0.69
10 (buffer) *	2.02 × 10^−2^	2.15 × 10^−4^	143.39	7.44 × 10^−3^	2.26	112.6	−11,704.0	0.69

**Table 4 nanomaterials-09-00745-t004:** Natamycin diffusion coefficients from Soluplus and Pluronic micelles, Soluplus:Pluronic P103 4:1 *v*/*v* mixed micelles, and poly(pseudo)rotaxanes in 0.9% NaCl and pH 6.4 buffer. Total copolymer concentration was 10% *w*/*v*. Mean values and, in parenthesis, standard deviation (n = 3).

Formulation	0.9% NaCl		Buffer pH 6.4	
	D × 10^6^ (cm^2^/min)	R^2^	D × 10^6^ (cm^2^/min)	R^2^
Pluronic P103 10%	49.46 (3.99)	0.992	65.14 (0.96)	0.992
Pluronic P103 + αCD 10%	32.66 (4.73)	0.990	50.30 (11.57)	0.990
Soluplus 10%	5.56 (1.62)	0.992	10.58 (0.53)	0.982
Soluplus 10% + αCD 10%	3.72 (0.24)	0.982	9.92 (2.38)	0.992
Soluplus/Pluronic P 103 (4:1)	18.31 (1.24)	0.998	45.68 (4.10)	0.989
Soluplus/Pluronic P 103 (4:1) + αCD 10%	8.21 (1.22)	0.970	16.14 (2.00)	0.996

**Table 5 nanomaterials-09-00745-t005:** Transcleral steady state flux (*J*) and permeability coefficients (*P_app_*) recorded for natamycin formulated in Soluplus and Pluronic micelles, Soluplus:Pluronic P103 4:1 *v*/*v* mixed micelles, and poly(pseudo)rotaxanes in 0.9% NaCl and pH 6.4 buffer. Total copolymer concentration was 10% *w*/*v*. Mean values and, in parenthesis, standard deviation (n = 3).

Formulation	0.9% NaCl		Buffer pH 6.4	
	*J* (µg/(cm^2^·h))	*P_app_* × 10^6^ (cm/s)	*J* (µg/(cm^2^·h))	*P_app_* × 10^6^ (cm/s)
Pluronic P103 10%	0.724 (0.042)	1.67 (0.09)	0.774 (0.032)	1.79 (0.07)
Pluronic P103 + αCD 10%	0.496 (0.023)	1.15 (0.05)	0.449 (0.012)	1.04 (0.03)
Soluplus 10%	0.403 (0.006)	0.93 (0.01)	0.466 (0.024)	1.08 (0.05)
Soluplus 10% + αCD 10%	0.174 (0.018)	0.40 (0.04)	0.090 (0.008)	0.27 (0.02)
Soluplus/Pluronic P 103 (4:1)	0.623 (0.065)	1.44 (0.15)	0.543 (0.021)	1.26 (0.05)
Soluplus/Pluronic P 103 (4:1) + αCD 10%	0.391 (0.023)	0.91 (0.05)	0.313 (0.016)	0.73 (0.04)

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
