# Peer review of "Cyclodextrin–Amphiphilic Copolymer Supramolecular Assemblies for the Ocular Delivery of Natamycin"

_nanomaterials, 2019, doi:10.3390/nano9050745_

Reviewer 1 Report

The article is very well written, easy to follow with conclusions supported by results. I do support its publication. Schemes of the different types of delivery systems could be added.

Author Response

Thank you very much for your positive comments and suggestions. We have now included the scheme of the different types of delivery systems in Supporting Information, Figure S1 as suggested.

Reviewer 2 Report

The paper deals with the controlled ocular delivery of Natamycin by formulations based on Soluplus or Pluronic with or without an alpha-cyclodextrin.

This paper has very clear objectives, is very well-written and is easy to read. I recommend its publications because my comments are really minor.

Soluplus is a commercial polymer. Nevertheless, I advise to add its composition because I only found the molar mass in the description of materials.

The authors describe the increase of solubility of Natamicyn in the presence of the copolymers. I advise to be more cautious with the vocabulary used. Indeed, strictly speaking, the solubility of the drug in water is not increased. Of course, the drug is loaded inside the core of micelles, which allow to increase the amount of drug in the formulation. This is just an increase of an apparent solubility. The word “apparent solubility” was used for instance in the title of figure 2 but was not used systematically within the text.

I have a similar remark on the sentence “The capability of CD to solubilize natamycin was verified”. The sentence is not 100% correct because the solubility of the drug is not increased but the drug is converted into a new chemical species, which is more soluble in water. For instance, I advise a sentence such as “the capability of CD to form an inclusion complex with natamycin, which is more soluble compared to uncomplexed natamycin.

I have a comment for lign 425, which accounts for faster diffusion by a size effect (smaller micelles). Of course, I agree that size is an important factor contributing to diffusion kinetics. Nevertheless, other factors could also be important such as interactions between drug and core of micelles, crystallinity, glass transition temperature…. Polymers used in this study having a different chemical structure, I was thus wondering if different interactions with the core could also influence diffusion kinetics.

Author Response

Thank you very much for your positive comments and suggestions. Soluplus composition is now described in more detail in the Introduction, line 71.

The term “apparent solubility” is now used along the manuscript.

The sentence on the capability of CDs to form inclusion complexes has been modified as suggested.

In section 3.4, we have now clarified that other factors may also contribute to the diffusion process

Reviewer 3 Report

The paper of Lorenzo-Veiga et al reports on the preparation and characterization of nanocarriers made of Pluronic 103, Soluplus, a mixture of them in the presence and in the absence of alfa-CD for a drug approved for fungal keratitis, tha is Natamycin.

Different techniques have been used for investigating these nanocarriers: laser light scattering for size determination and zeta-potential, spectrophotometry for solubility measurements, rheological characterization, diffusion assays, ocular tolerance and ex-vivo cornea and sclera permeability.

The subject of the manuscript is interesting and deserve a deep investigation, nevertheless the present manuscript needs to be strongly revised in order to clarify some relevant aspects. In particular, some of the conclusions are not always supported by the results.

The manuscript is well written.

The following points should be considered in order for the manuscript to be publishable.

1) Page 4, line 139: it is not clear if X is the capital for a symbol type or for a normal type. I realize that it is the capital for a symbol type by reading Tables 2 and 3. Please correct it.

2) Page 4, line 146: The definition of PM is not very clear to me. Why don’t they use a simpler definition such as X/Sw? In the definition of PM the authors insert an arbitrary default concentration of 1 M and consider the CMC of the polymer twice, in X and in the numerator.

3) Page 4, line 157: “buffer and 0.9% NaCl” should be changed to “buffer or 0.9% NaCl”.

4) Page 5, line 202: I think that, in order for readers to clearly understand the ocular tolerance test, it is important to clearly define haemorrhage, vascular lysis and coagulation and the way to evaluate and differentiate them.

5) Page 7, Figure 2. Why did the Authors not insert in the Figure the most investigated 10% concentration of copolymers?

6) Page 8, line 287-289. Authors should clarify how do they evaluate the concentration of NAT (Tables 2 and 3) in Soluplus or mixed Soluplus/P103 micelles as they said that Soluplus adsorbs light at the same 304 nm wavelength.

7) As far as the stability of micelles is concerned, the data referring to non-diluted sample should be inserted in Figure S1.

8) Page 8, Tables 2 and 3 are very informative. Why did the Authors not insert a similar Table for the mixed copolymer micelles (at least the most investigated 4:1)? I think a similar Table could be very useful.

9) Page 8, Tables 2 and 3. Since in the text Authors refer to concentration in terms of ug/mL, I would advice them to insert a further column in which NAT concentration is expressed in terms of ug/mL.

10) Page 9, lines 316-318. Authors claim that the apparent CMC of mixed micelles varies following changes of micelles dimensions. It would be interesting to calculate the CMC of mixed micelles of all ratios in order to confirm this point.

11) Page 11, line 386-387: “Soluplus/P103 4:1 dispersions showed a shift in the sol-gel transition towards higher values compared to Soluplus solely dispersion”. This sentence is not completely true because in pure unloaded system in NaCl 0.9% this is not true.

12) Page 12, lines 411-413. In order to understand the calculation of IS, I suggest to insert in the Supporting Information a Table with tH, tL and tC values of different systems.

13) Page 12, lines 425-427. I’m not so sure that the diffusion findings are in agreement with the smaller size of Pluronic P103 micelles. What about the amount of NAT solubilized in water? It is much higher in the P103 micelles compared to Soluplus system. I cannot compare the other mixed systems because I have no data. For this reason I warmly suggest Authors to insert Tables similar to 2 and 3 for the mixed system (comment 6) and the ones containing alfa-CD (at least at 10% CD). Moreover, how do these results compare with the almost absence of variation following a sudden 30- or 60-fold dilution (Figure S1)? Does the drug leave the micelles or keep its solubilization in the micellar core?

14) Page 12, Figure 6. In order to compare the amount of NAT loaded in the micelles and that diffused, it would be useful to insert in the graphs of Figure 6, in the y-axis on the right, the NAT concentration in terms of ug/mL.

15) Page 12, line 434: amend “nantamycin”.

16) Since the ex-vivo experiments required a carbonate buffer of 7.2, why did the Author use a buffer of 6.4 for the previous experiments?

17) Page 14, line 481 and following. It is not clear if Authors hypothesize that drug, in ex vivo experiments, leaves the micelles interacting with sclera and cornea or not. It is important to highlight this point.

Author Response

We are very grateful to the reviewer for his/her detailed comments and suggestions. We have answered as follows.

The following points should be considered in order for the manuscript to be publishable.

1)Page 4, line 139: it is not clear if X is the capital for a symbol type or for a normal type. I realize that it is the capital for a symbol type by reading Tables 2 and 3. Please correct it.

RESPONSE: The symbol for molar solubilization capacity has been revised accordingly.

2) Page 4, line 146: The definition of PM is not very clear to me. Why don’t they use a simpler definition such as X/Sw? In the definition of PM the authors insert an arbitrary default concentration of 1 M and consider the CMC of the polymer twice, in X and in the numerator.

RESPONSE:  Thank you for the comment. Equation 3 is commonly used in papers devoted to drug solubilization and thus, for comparison purposes, we prefer to use this definition of PM.

3) Page 4, line 157: “buffer and 0.9% NaCl” should be changed to “buffer or 0.9% NaCl”.

RESPONSE: The sentence on line 158 (former 157) has been modified accordingly.

4) Page 5, line 202: I think that, in order for readers to clearly understand the ocular tolerance test, it is important to clearly define haemorrhage, vascular lysis and coagulation and the way to evaluate and differentiate them.

RESPONSE: The terms haemorrhage, vascular lysis and coagulation are now described in lines 203-204. Thank you

5) Page 7, Figure 2. Why did the Authors not insert in the Figure the most investigated 10% concentration of copolymers?

RESPONSE: Data for 10% copolymer are shown in Figure 3.

6) Page 8, line 287-289. Authors should clarify how do they evaluate the concentration of NAT (Tables 2 and 3) in Soluplus or mixed Soluplus/P103 micelles as they said that Soluplus adsorbs light at the same 304 nm wavelength.

RESPONSE:  This low Soluplus UV absorption did not interfere with drug solubility quantification in the apparent solubility studies as the micelles disassemble in the ethanol-water medium, but it caused noise under the dilution with aqueous medium of the stability test. This is now clarified in lines 291-294.

7) As far as the stability of micelles is concerned, the data referring to non-diluted sample should be inserted in Figure S1.

RESPONSE: The data for the non-diluted Pluronic samples are already given in Figure 3.

8) Page 8, Tables 2 and 3 are very informative. Why did the Authors not insert a similar Table for the mixed copolymer micelles (at least the most investigated 4:1)? I think a similar Table could be very useful.

RESPONSE: Thank you for the comment. We have now added the partition coefficient, P, values for Soluplus /Pluronic P103 1:4 and 4:1 (vol/vol) systems in lines 318-319.

9) Page 8, Tables 2 and 3. Since in the text Authors refer to concentration in terms of ug/mL, I would advice them to insert a further column in which NAT concentration is expressed in terms of ug/mL.

RESPONSE: Tables 2 and 3 have been modified to show the NAT concentration in mg/mL.

10) Page 9, lines 316-318. Authors claim that the apparent CMC of mixed micelles varies following changes of micelles dimensions. It would be interesting to calculate the CMC of mixed micelles of all ratios in order to confirm this point.

RESPONSE: We agree with the comment, but the manuscript focused on the usefulness of the micelles on NAT solubilization. A detailed analysis of the CMC of mixed micelles would involve strong research efforts beyond the scope of our study.

11) Page 11, line 386-387: “Soluplus/P103 4:1 dispersions showed a shift in the sol-gel transition towards higher values compared to Soluplus solely dispersion”. This sentence is not completely true because in pure unloaded system in NaCl 0.9% this is not true.

RESPONSE: The sentence on the sol-gel transition has been revised accordingly (lines 396-397).

12) Page 12, lines 411-413. In order to understand the calculation of IS, I suggest to insert in the Supporting Information a Table with tH, tL and tC values of different systems.

RESPONSE: The IS was 0.0 for all formulations; namely none micelle or poly(pseudo)rotaxane dispersions caused hemorrhage, lysis, or coagulation. Therefore, tH, tL and tC values were in all cases bigger than 301 seconds. This is now explained in section 3.3.

13) Page 12, lines 425-427. I’m not so sure that the diffusion findings are in agreement with the smaller size of Pluronic P103 micelles. What about the amount of NAT solubilized in water? It is much higher in the P103 micelles compared to Soluplus system. I cannot compare the other mixed systems because I have no data. For this reason I warmly suggest Authors to insert Tables similar to 2 and 3 for the mixed system (comment 6) and the ones containing alfa-CD (at least at 10% CD). Moreover, how do these results compare with the almost absence of variation following a sudden 30- or 60-fold dilution (Figure S1)? Does the drug leave the micelles or keep its solubilization in the micellar core?

RESPONSE: Thank you for the comment. Following the referee suggestions, We have now added the partition coefficient, P, values for Soluplus /Pluronic P103 1:4 and 4:1 (vol/vol) systems in lines 318-319. The partition coefficients P for Soluplus /Pluronic P103 4:1 (vol/vol) were in the 1.72-1.82 range. These values are lower than those recorded for pure Soluplus or Pluronic in separate. This means that there are more free drug in the mixed micelles. Nevertheless, the diffusion is slower than from Pluronic micelles. Therefore, our hypothesis is that the smaller size of Pluronic micelles plays a relevant role in the diffusion rate, although other factors such as copolymer composition and drug-core interactions may also affect to the diffusion kinetics. This is now explained in more detail in lines 438-444.

14) Page 12, Figure 6. In order to compare the amount of NAT loaded in the micelles and that diffused, it would be useful to insert in the graphs of Figure 6, in the y-axis on the right, the NAT concentration in terms of ug/mL.

RESPONSE: All formulations were loaded with the same amount of NAT. Therefore, starting concentrations for all systems were the same. Since the volume of the diffusion cells was also the same for all test, the plot would show the same pattern either as ug/cm2 or ug/mL.

15) Page 12, line 434: amend “nantamycin”.

RESPONSE: It has been modified. Thank you.

16) Since the ex-vivo experiments required a carbonate buffer of 7.2, why did the Author use a buffer of 6.4 for the previous experiments?

RESPONSE: Ex vivo experiments we carried using carbonate buffer pH 7.2 as receptor medium as recommended in the BCOP test protocol. Thus buffer pH 7.2 was only used as receptor medium. The formulations were prepared as for the other experiments, either in 0.9% NaCl or buffer pH 6.4. This is now explained in more detail in lines 216-217.

17) Page 14, line 481 and following. It is not clear if Authors hypothesize that drug, in ex vivo experiments, leaves the micelles interacting with sclera and cornea or not. It is important to highlight this point.

RESPONSE: Thank you for the comment. We hypothesize that to cross the cornea, the drug is expected to abandon the micelles/CDs. Differently, micelles can cross sclera due to the larger pores of this tissue. This is now clarified in lines 513-514.

Round  2

Reviewer 3 Report

I think that the Authors answered and amended almost all the points highlighted by this Referee.

Nevertheless, the following points need further clarification:

Previous revision: 7) As far as the stability of micelles is concerned, the data referring to non-diluted sample should be inserted in Figure S1.

RESPONSE: The data for the non-diluted Pluronic samples are already given in Figure 3.

Comment 1: I think that Figure 1S (2S in the revised version of Supplementary) reports different data with respect to Figure 3. I think that, for comparison reason, and in order to understand the stability of the non-diluted Pluronic sample it is important to add the stability data over time of this sample in Figure 2S.

Previous revision: 10) Page 9, lines 316-318. Authors claim that the apparent CMC of mixed micelles varies following changes of micelles dimensions. It would be interesting to calculate the CMC of mixed micelles of all ratios in order to confirm this point.

RESPONSE: We agree with the comment, but the manuscript focused on the usefulness of the micelles on NAT solubilization. A detailed analysis of the CMC of mixed micelles would involve strong research efforts beyond the scope of our study.

Comment 2: I think that, although the manuscript is devoted to the solubilization of NAT in micelles, the Authors should not claim something that they did not prove. Therefore, if they do not measure the apparent CMC of mixed micelles, it would be proper not to mention to these, not determined, data in the comment. Perhaps they could refer to literature data.

Author Response

Comment 1: I think that Figure 1S (2S in the revised version of Supplementary) reports different data with respect to Figure 3. I think that, for comparison reason, and in order to understand the stability of the non-diluted Pluronic sample it is important to add the stability data over time of this sample in Figure 2S.

Answer: We have now modified Figure S2, and the plots of non-diluted Pluronic samples were added. The caption has been modified accordingly.

Comment 2: I think that, although the manuscript is devoted to the solubilization of NAT in micelles, the Authors should not claim something that they did not prove. Therefore, if they do not measure the apparent CMC of mixed micelles, it would be proper not to mention to these, not determined, data in the comment. Perhaps they could refer to literature data.

Answer: Thank you for the advice. We have modified the text accordingly and no reference to the apparent CMC is now made.